# Functional interpretation of single cell similarity maps

David DeTomaso ⬚ [1,7], Matthew G. Jones[2,7], Meena Subramaniam[2], Tal Ashuach[1], Chun J. Ye ⬚ [3] & Nir Yosef ⬚ [4,5,6]

We present *Vision*, a tool for annotating the sources of variation in single cell RNA-seq data in an automated and scalable manner. *Vision* operates directly on the manifold of cell-cell similarity and employs a flexible annotation approach that can operate either with or without preconceived stratification of the cells into groups or along a continuum. We demonstrate the utility of *Vision* in several case studies and show that it can derive important sources of cellular variation and link them to experimental meta-data even with relatively homogeneous sets of cells. *Vision* produces an interactive, low latency and feature rich web-based report that can be easily shared among researchers, thus facilitating data dissemination and collaboration.

[1] Center for Computational Biology, University of California Berkeley, Berkeley, CA, USA. [2] Biological and Medical Informatics Graduate Program, University of California, San Francisco, CA, USA. [3] Department of Epidemiology and Biostatistics, Department of Bioengineering and Therapeutic Sciences, Institute for Human Genetics, University of California, San Francisco, CA, USA. [4] Department of Electrical Engineering and Computer Science and Center for Computational Biology, University of California, Berkeley, Berkeley, CA, USA. [5] Ragon Institute of Massachusetts General Hospital, MIT and Harvard, Cambridge, MA, USA. [6] Chan-Zuckerberg Biohub, San Francisco, CA 94158, USA. [7] These authors contributed equally: David DeTomaso, Mathew G. Jones. Correspondence and requests for materials should be addressed to N.Y. (email: niryosef@berkeley.edu)

Recent technological advancements have allowed transcriptional profiling at the single-cell level[1–3]. This has enabled a deeper investigation into cellular heterogeneity[4], the identification of new cellular subtypes[5,6], and more detailed modeling of developmental processes[7,8]. Notably, the data produced in a single-cell RNA-seq (scRNA-seq) experiment is distinct from that of bulk RNA-seq in that it is typically sparse (with many expressed genes remaining undetected), and consists of a very high number of data points[9]. Furthermore, most scRNA-seq studies encompass cells of different types or states in one sample, without preconceived labeling of these cells into phenotypic groups.

A typical primary step in the analysis of scRNA-seq data is therefore to extract a meaningful labeling by partitioning the cells into clusters[10,11] or by placing the cells along some continuum[12] in a data-driven manner. A common way to achieve this is to first project the data onto a low-dimensional space, which preserves critical information while reducing noise and (depending on the method) bias. While principal component analysis (PCA) is a commonly used projection method, more recently linear factor models, such as ZIFA[13] or ZINB-WaVE[14] and nonlinear deep generative models, such as scVI[11] or DCA[15] have been developed to specifically address the underlying distributions and confounders found in single-cell RNA-sequencing. The resulting manifold representations[16] can then be used as a baseline for dividing the cells into clusters. Alternatively, if the cells are expected to vary along a continuum, such as that which arises during a developmental time-course, a tree-like representation of the data can be inferred instead, based on the same manifold (summarized in ref. [12]).

While the assignment of labels (e.g., cluster IDs) to cells greatly simplifies the interpretation of the data, it may come at the cost of missing important yet subtle patterns of variation (e.g., gradients of important cellular functions within a cluster of cells[17]) and suffer from inaccuracies (e.g., when there is no obvious cluster structure[18]). Furthermore, even once labels have been assigned, it may still not be clear how to interpret their underlying biological meaning. To address these challenges and identify the key biological properties that dominate the variability between cells in a sample, we developed *Vision*: a flexible annotation tool that can operate both with and without a preconceived labeling of cells (Fig. 1a). As an input *Vision* takes a map of similarities between cells, which can be computed internally or provided by external manifold learning algorithms[11,13,14,16]. *Vision* then leverages the concept of transcriptional signatures[17,19] to interpret the meaning of the variability captured in the manifold by integrating information from a large cohort of published genome-scale mRNA profiling datasets[20–22]. In its label-free mode, *Vision* operates directly on the single cell manifold and uses an autocorrelation statistic to identify biological properties that distinguish between different areas of the manifold. The result is a set of labelings of the cells which may differ when studying different aspects of cell state (e.g., tissue context vs. differentiation stage in T cells[18]). This approach is therefore capable of highlighting numerous gradients or sub-clusters which reflect varied cellular functions or states and which may not be captured by a single stratification of the cells into groups. As we demonstrate, this approach is particularly helpful when studying cells from a similar type (e.g., T helper cells), with no clear partitioning. In its label-based mode, *Vision* identifies biological properties that differ between precomputed stratifications (e.g., clusters) or that change smoothly along a given cellular trajectory. To enable the latter, *Vision* utilizes the API built by Saelens and colleagues[12] to support a large number of trajectory inference methods, and to our knowledge it is the first functional-annotation tool to do so.

*Vision* has several additional properties that distinguish it from other software packages for automated annotation and for visualization and exploration of single cell-data (summarized in Supplementary Table 1). Foremost, *Vision* is designed to naturally operate inside of analysis pipelines, where it fits downstream of any method for manifold learning, clustering, or trajectory inference and provides functional interpretation of their output. Indeed, in the following we demonstrate the use of *Vision* within three different pipelines consisting of stratification free analysis where similarity between cells is based on either PCA or scVI, and stratification-based analysis where cells are organized along a developmental pseudo-time course.

As we further demonstrate with these case studies, *Vision* also enables the exploration of the transcriptional effects of meta-data, including cell-level (e.g., technical quality or protein abundance[23]) and sample-level (e.g., donor characteristics) properties. Finally, the use of *Vision* can greatly facilitate collaborative projects, as it offers a low-latency report that allows the end-user to visualize and explore the data and its annotations interactively. The report can be hosted on-line and viewed on any web browser without the need for installing specialized software (Fig. 1b). *Vision* is freely available as an R package at www.github.com/YosefLab/VISION.

## Results

**Using signature scores to interpret neighborhood graphs.** *Vision* operates on a low-dimensional representation of the transcriptional data and starts by identifying, for each cell, its closest $K$-nearest neighbors in the respective manifold. Computing this for every cell results in a $K$-nearest-neighbor (KNN) graph. By default, *Vision* performs PCA to create this low-dimensional space, but the results of more advanced latent space models[11,13,14] or trajectory models (via[12]) can be provided as an input instead (to note, these trajectory models may be described as both latent spaces and a precomputed labeling of the cells). In order to interpret the variation captured by the KNN graph, *Vision* makes use of gene signatures—namely, manually annotated sets of genes, which describe known biological processes[24] or data-driven sets of genes that capture genome-wide transcriptional differences between conditions of interest[25]. These signatures are available through databases, such as MSigDB[26], CREEDS[21], or DSigDB[22] and can also be assembled in a project-specific manner (e.g., as in refs. [17,27]). For each signature, an overall score is computed for every cell summarizing the expression of genes in the signature. For example, with a signature describing inflammatory response, a high signature score would indicate that the cell in general has higher expression of known inflammatory response genes. Gene signatures may also be 'signed'—representing a contrast between two biological conditions. For example, given a signature representing Th17 helper cells vs. regulatory T cells, a higher score would indicate that a cell's transcriptional program is more Th17-like, while a lower score would indicate it is more similar to the regulatory state[17,28]. To reduce the effect of batch or technical covariates on these signature scores, we recommend the use of a normalization procedure (such as in refs. [29,][11] or [30]) on the gene expression dataset prior to input into *Vision*.

To interpret the cell–cell KNN graph in the context of signature scores without the use of labels (namely, label-free mode), we make use of a local autocorrelation statistic, the Geary's $C$[31]. This statistic was originally developed for use in demographic analysis to identify significant spatial associations (e.g. "Are incident rates of obesity randomly distributed within a city or is there a certain areal pattern?"). Here, we make use of this statistic to answer a similar question: "Is the signature score

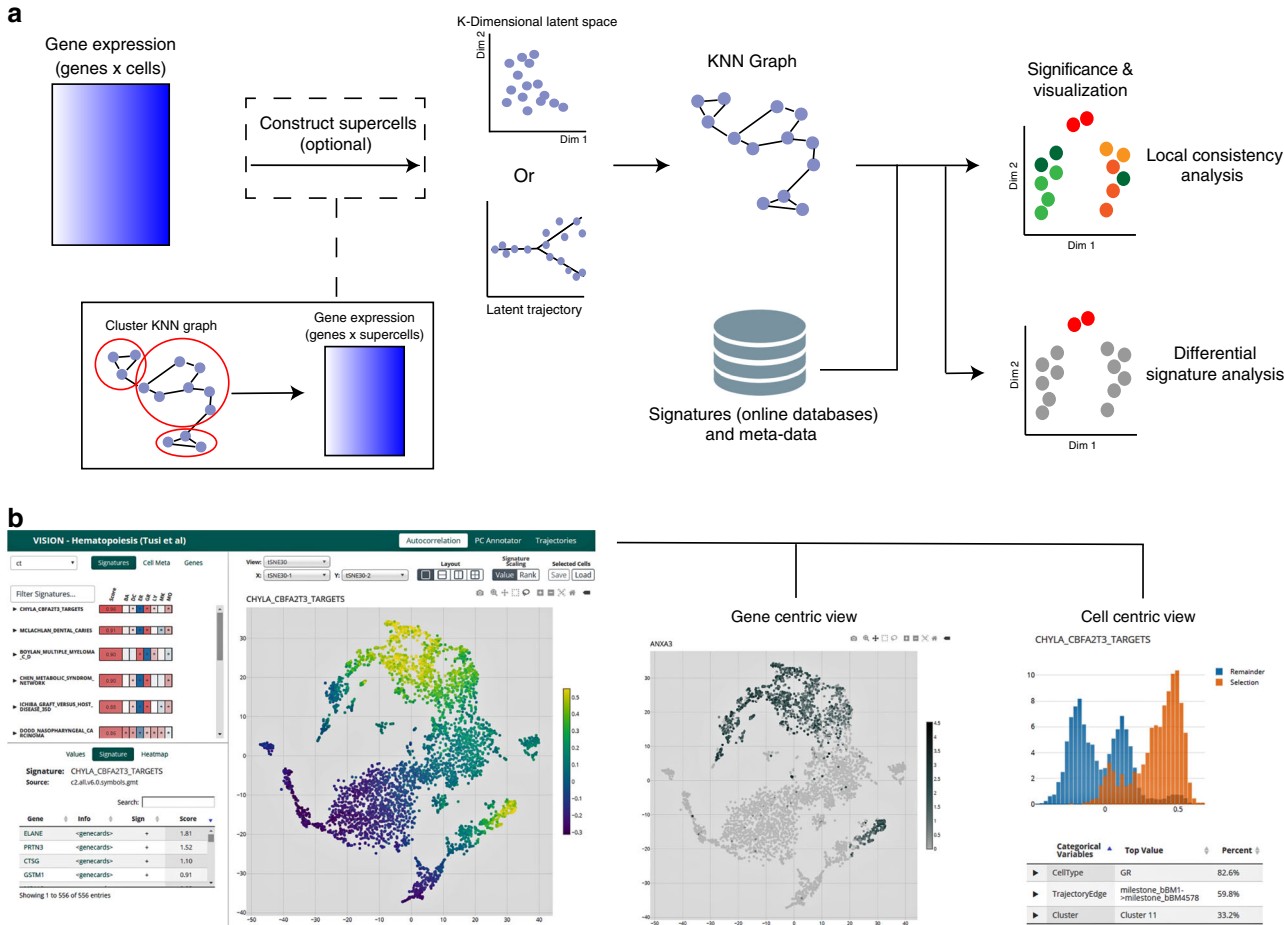

**Fig. 1** *Vision* is a dynamic framework for annotating and exploring scRNA-seq datasets with a high-throughput pipeline and interactive, web-based report. **a** The *Vision* processing pipeline consists of several key steps. A *K*-nearest-neighbors graph is constructed within the latent model for gene expression (either supplied as input, or computed using PCA). Optionally, this graph can be reduced for more efficient downstream computation by combining similar cells into 'micro-clusters'. Representative scores are evaluated for external gene signatures and local autocorrelation analysis is used to evaluate which signatures best reflect the heterogeneity in the latent space. **b** The analysis output is organized into an interactive report in which signatures, individual gene expression, and cell meta-data can be visualized along with two-dimensional representations of the data

randomly distributed within the KNN graph or are there certain groups of neighboring cells with values higher or lower than expected?" We report $C' = 1 -$ Geary's C as the effect size so that a value of 0 indicates no local autocorrelation and a value of 1 corresponds with maximal autocorrelation. To assess significance, an empirical *p*-value is computed by comparing the reported $C'$ value to a distribution drawn from randomly generated signatures (see the "Methods" section for a complete description of this statistical test).

In this manner, signatures with a significantly high local autocorrelation statistic can then be used to assign a biological meaning to specific areas of the KNN graph, and also capture gradients or various sub-divisions along the graph. This is especially useful for cases where the cells do not clearly separate into discrete clusters, but rather exhibit variation in a more continuous fashion. Importantly, we demonstrate that this statistic is stable with respect to the number of neighbors taken to form the KNN graph (Supplementary Fig. 1). In its label-based mode *Vision* evaluates the dependence of the biological signatures on the labels assigned to each cell, such as experimental group or cluster ID. To accomplish this, *Vision* uses a 1-vs.-all differential signature test (using the Wilcoxon rank-sum test and the associated area under the ROC curve statistic, AUROC; see the

"Methods" section) to highlight biological properties that distinguish each cluster.

In addition to gene signatures, *Vision* allows the user to directly input other quantitative properties (meta-data) for each cell (e.g., protein abundance[23]) or each sample (e.g., donor clinical parameters), and explore their effects in a similar manner to that of the gene signatures. The meta-data may also be categorical and represent properties, such as batch annotations or specifications of the respective experimental condition. *Vision* enables the analysis of these categorical properties in both local autocorrelation (label-free) and comparative (label-based) modes using the chi-squared test and the associated Cramér's V statistic (see the "Methods" section). This approach can highlight important relationships between data and meta-data (e.g., identifying signatures associated with a certain experimental condition) and also evaluate the degree to which quality control (QC) parameters (e.g., % of mapped reads per cell) and other potential confounding factors (e.g., batch ID) may influence the observed variability between cells.

**Uncovering cellular responses to chemical perturbations**. As a demonstration, we applied *Vision* to a published scRNA-seq dataset consisting of 29,000 PBMCs from eight Lupus patients[32].

Samples were further divided into unstimulated and interferon-beta stimulated batches prior to scRNA-seq. To infer a latent space for cell–cell similarities, PCA was used on the log-transformed scaled counts. As expected, when analyzing the full dataset consisting of all PBMCs, the structure of the data is predominantly defined by cell type differences. The signature scores of cell type-specific signatures reflect this stratification (Fig. 2a). When further examining just the CD4 T Cells, an unsupervised analysis reveals that the activation of interferon-response genes is a major driver of cell–cell variation. Signatures with the highest local autocorrelation include interferon alpha/gamma response signatures from MSigDB[20] and an interferon-beta response signature ($C' = 0.73$, FDR<$2.4\times 10^{-3}$) which we added from ref. [33] as a positive control (Fig. 2a). Alternately, stratifying the cells based on known cell meta data (stimulated vs. unstimulated) and running differential signature analysis highlights the interferon-beta signature as the top result (AUROC = 0.99, FDR<$1\times 10^{-16}$).

We then subset the data further to investigate more fine-grained variation within the interferon-stimulated CD4 T cells. Signatures with significant local autocorrelation ($C' > 0.2$ and FDR<0.05) were clustered based on their scores across cells, so that different patterns of variation could be identified and paired with functional annotations (Fig. 2b). The largest group of signatures identified in this manner consists of signatures that distinguish naive and memory T cells (such as that from ref. [34] via MSigDB) and broadly divide the stimulated T cells into two large clusters (Fig. 2d). A second component driving variation consists of interferon response signatures, which further stratify the cells within both the naive and memory clusters (Fig. 2c). As all of these cells were stimulated by interferon-beta, this likely represents variable activation of the interferon response pathway among stimulated cells. Notably, and in contrast to the naive-memory component, the interferon response signatures do not have high autocorrelation within the unstimulated CD4 T cells (Supplementary Fig. 2a), demonstrating that this variation likely occurs as a direct consequence of interferon-stimulation and is not significantly present among unperturbed human CD4 T cells. Additionally, this variation does not appear to be a consequence of donor–donor differences as the autocorrelation is removed under within-donor permutations of the signature scores ($p<.001$, Supplementary Fig. 2b). Lastly, a third identified component of variation contains signatures, such as antigen processing and presentation (KEGG[24]) and is characterized by an increase in MHC (class I and II) expression (Supplementary Fig. 2c). Since upregulation of MHC class II transcripts has been observed in human CD4 T cells as a consequence of prolonged stimulation[35], this signature may be indicating a subset of CD4 T cells undergoing long-term activation.

In this way, Vision is used to analyze a large, diverse sample by first describing the differences between major cell clusters and then further annotating the more nuanced biological variation within individual clusters.

**Identifying myeloid-specific programs in AML.** To further demonstrate the ability of Vision to detect relevant stratifications in large scRNA-seq datasets, we turned to a collection of 38,410 cells from bone marrow aspirates of 16 patients with acute myeloid leukemia (AML) and five healthy donors[36]. In this analysis, we modeled the latent space using scVI[11], an alternative method to PCA for providing normalized, low-dimensional spaces via non-linear transformation. Additionally, we leverage information beyond gene expression and include cell type labels that were inferred in the original study (which can be used for label-based analysis), as well as donor IDs and disease status (provided as categorical meta-data).

While scVI has been applied so as to correct for technical batch effects (each batch corresponding to a donor), it may still not force strict alignment between datasets that are biologically different[37]. Indeed, we find in the full dataset consisting of all cells that the patient IDs drive much of the variation ($V = 0.60$, FDR<$3.1\times 10^{-3}$; Fig. 3), in addition to the provided stratification into cell types ($V = 0.59$, FDR<$3.1\times 10^{-3}$). As noted in the original study, this donor effect may be due to varying cell type proportions per donor, or indicate that AML can progress in one of several ways depending on patient-specific driver mutations. Regardless of cause, Vision complements other recent work, in that it provides a natural and quantitative framework for assessing the effect of these categorical (and possibly nuisance) factors[38].

We hypothesized that we would be able to better dissect the myeloid-specific behavior in AML patients by analyzing a specific subpopulation composed of monocyte and monocyte-like cells (Fig. 3). In the original study, the authors find that AML patients had far fewer effector T cells and suggest that a subset of differentiated CD14$^+$ monocyte-like cells may play an immuno-suppressive role. In our selected subset of 7280 cells we still find that patient ID, and more generally the disease status are significant sources of variation ($V = 0.52$, FDR<$2.6\times 10^{-3}$ and $V = 0.75$, FDR<$2.6\times 10^{-3}$, respectively, Supplementary Fig. 3a). However, we also find signatures that corroborate the findings of the original study. Specifically, we find two major axes that summarize the dataset: First, an immunosuppressive axis consisting of pro-tumor, tumor-associated macrophage (TAM) cell markers and characterized by the Stearman tumor field effect signature ($C' = 0.51$, FDR<0.01, Fig. 3c), which is up-regulated in the AML myeloid cells (FDR<$1\times 10^{-16}$, Supplementary Fig. 3e). Other signatures support this observation that the myeloid lineage in AML patients transitions towards an alternatively activated state, resembling an immunosuppressive M2-like macrophage population (GSE25088 WT vs. Stat6 KO Macrophage IL4 Stimulation, $C' = 0.7$, FDR<$2.7\times 10^{-3}$, Supplementary Fig. 3c). Second, we find that AML myeloid cells are less progenitor-like and that disease and healthy myeloid cells can be stratified along an axis of myeloid cell maturity, as demonstrated by the Eppert Progenitor signature ($C' = 0.49$, FDR<$2.7\times 10^{-3}$, Supplementary Fig. 3d). Both of these axes support the original findings that mature, differentiated myeloid cells help suppress T cell effector function in AML. On one hand, the loss of cytotoxic T cells observed by the authors is reflected by the development of an immunosuppressive micro-environment niche in the bone marrow (as measured with the Stearman tumor field effect signature). On the other hand, we find that this immunosuppressive niche is also characterized by an abundance of more differentiated myeloid cells expressing more mature myeloid markers, compared to the healthy bone marrow samples, such as CD11b (FDR<0.05, wilcoxon rank-sums test) and CD11c (FDR<$1.0\times 10^{-8}$ Wilcoxon rank-sums test).

Interestingly, one of the genes that comprise the Stearman tumor field effect signature is the enzyme SAT1, which as expected is also up-regulated in the differentiated, immunosuppressive monocytes that are abundant in the disease cohort ($p<1.0\times 10^{-8}$, Wilcoxon rank-sums test, Fig. 3c). SAT1 up-regulation is associated with increased activity of polyamine metabolism and thus higher concentration of intracellular polyamines. Polyamine production, in turn, has been generally shown to decrease tumor cytoxocity while increasing tumor cell proliferative ability[39]. Recent work has further indicated that increased polyamine metabolism in macrophages exposed to lipopolysaccharide (LPS) facilitates type-2 macrophage (M2)

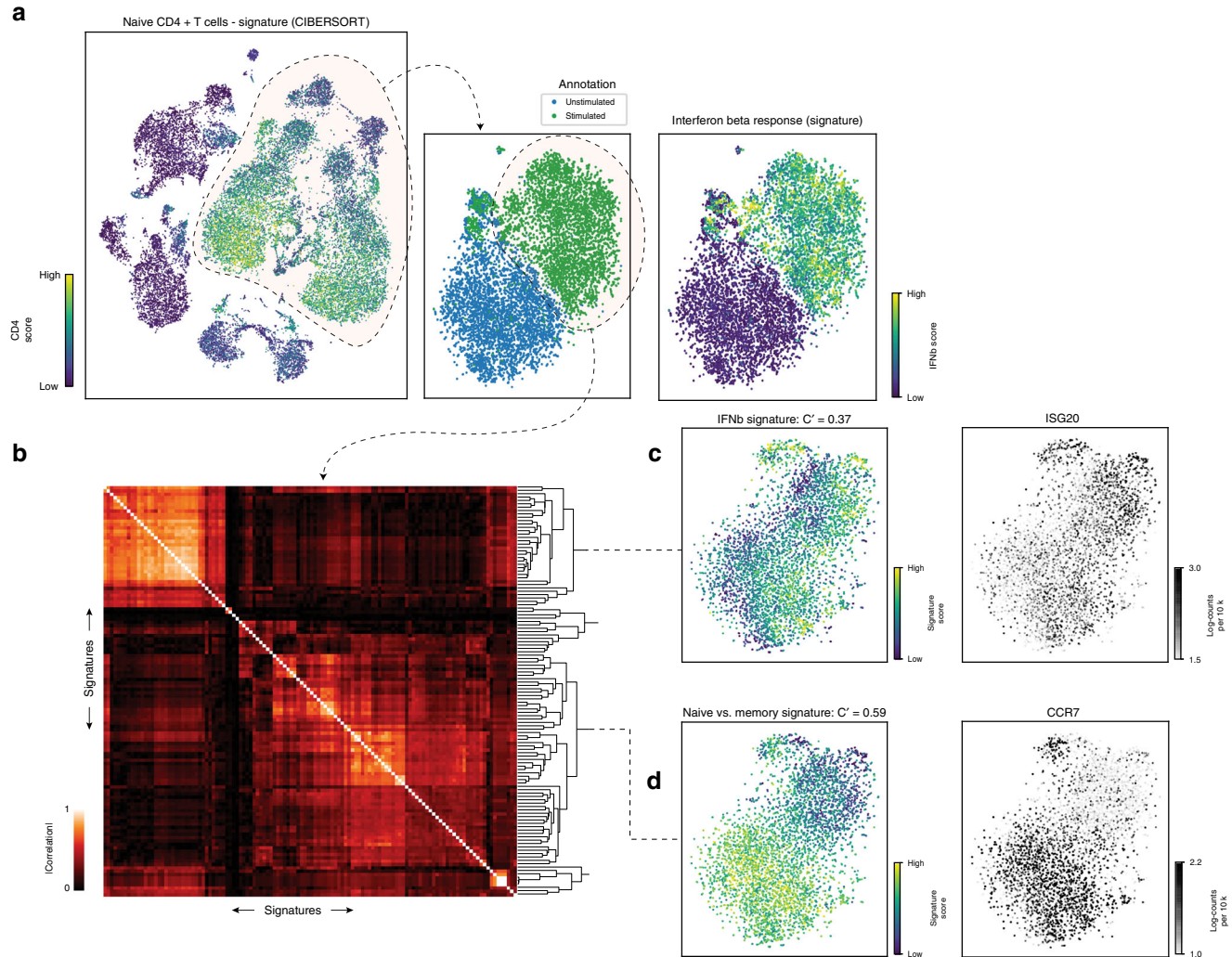

**Fig. 2** *Vision* highlights within-group and between-group variation within Lupus PBMCs. **a** Stimulated and unstimulated PBMCs from all eight donors. A cell type signature (from CiberSort[52]) is used to identify the CD4 T cell cluster. Isolating the CD4 T cells reveals that interferon beta stimulation is the dominant component of variation. This is identified through the use of an interferon-beta stimulation signature and confirmed with the cell annotations. **b** Further isolating only the stimulated CD4 T cells, signatures with significant local autocorrelation are clustered based on the absolute correlation of their per-cell signature scores to reveal signature modules. **c** One such module describes the naive vs. memory axis of variation (signature from ref. [34] via MSigDB, and naive T cell marker gene CCR7 shown). **d** A second module describes variable activation of the interferon response pathway within the stimulated CD4 cells (signature from ref. [33] and interferon-induced gene ISG20 shown)

polarization while suppressing LPS signaling and type-1 macrophage (M1) activation[40]. Consistently, we observe that signatures associated with M2 polarization are highly autocorrelated (GSE25088 WT vs. Stat6 KO macrophage IL4 stimulation $C' = 0.7$, FDR$<2.7 \times 10^{-3}$, Supplementary Fig. 3c) and upregulated in the patient samples ($p<1.0 \times 10^{-16}$ *Vision* differential signature test; Supplementary Fig. 3e).

Taken together, these findings raise the hypothesis that the monocyte population in AML patients may go through a metabolic shift that contributes to the alterations in their functionality and subsequent loss of effector T cells in the AML microenvironment. More broadly, these results demonstrate how a user may rapidly sift through a large dataset and characterize the biological processes and individual genes that may be associated or even contribute to a phenotype of interest.

**Annotating cellular trajectories during hematopoiesis.** Our framework for biological interpretation with local autocorrelation can also be applied to cell-labelings from trajectory maps, asking:

"Is there an association between the position of a cell in the trajectory and a certain biological function?" Similarly to how this is accomplished with latent spaces, *Vision* computes a KNN graph from pre-computed trajectory models, connecting cells that are close to each other in the inferred continuum. The autocorrelation statistic is then computed on the KNN graph in a manner similar to the analysis above. Importantly, *Vision* supports a variety of trajectory inference methods through integration with the Dynverse[12] package, which provides wrappers for over 50 different algorithms.

To demonstrate the utility of this module, we used Monocle2[7] to infer a differentiation trajectory on 5432 hematopoietic progenitor cells (HPCs) isolated from the bone marrow of adult mice[41]. *Vision*'s rendering of the Monocle output recapitulates the pattern of differentiation and the stratification of cells discussed in the original study—namely, an undifferentiated core giving rise to differentiated cells, most notably erythrocytes and granulocytes (Supplementary Fig. 4). We used the Hallmark (H) and curated gene set (C2) collections from MSigDB[26] to attribute additional meaning to the differentiation process. Unsurprisingly,

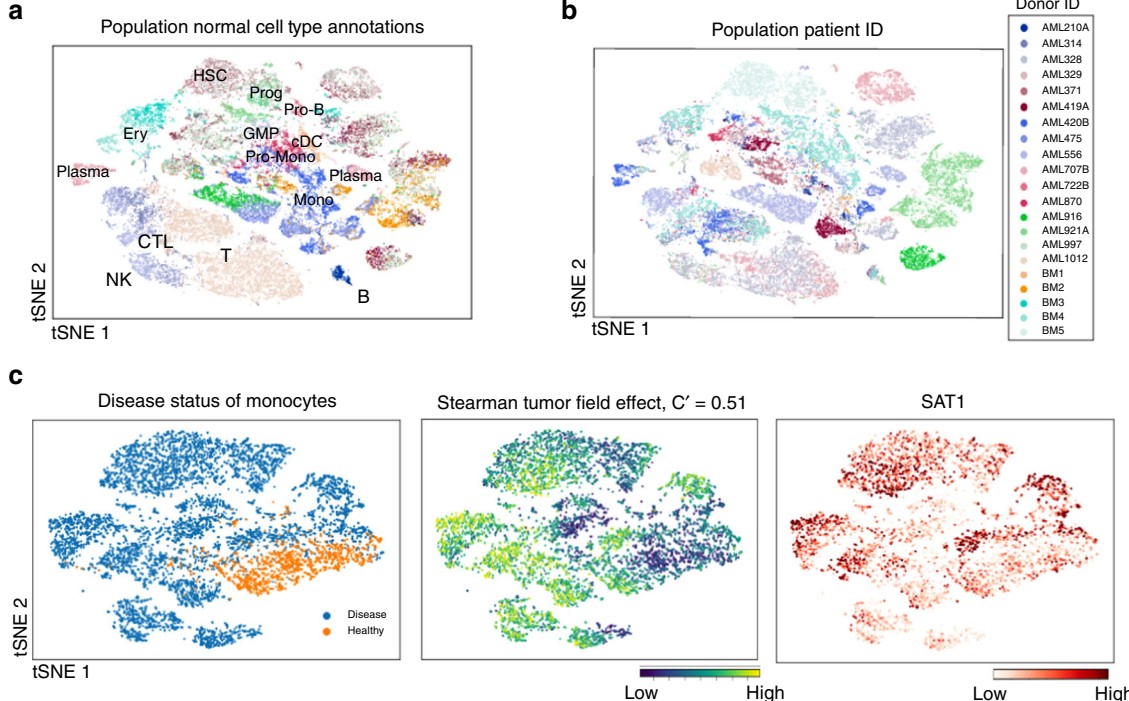

**Fig. 3** *Vision* identifies a metabolic switch associated with immunosuppressive monocytes in AML. **a** and **b** At the population level of 38,410 cells, major transcriptomic differences are captured by either the precomputed cell-type stratification **a** or donor ID **b**. Major cell types identified in the original study are annotated in **a** with the following abbreviations: NK natural killer, CTL cytotoxic T, Ery erythrocyte, HSC hematopoeitic stem cell, Prog progenitor, GMP granulocyte–monocyte progenitor, Mono monocyte. **c** In a subset of 7780 monocyte and monocyte-like cells, we find signatures associated with immunosuppression to have high values in the cells from disease patients. This is manifested in the Stearman tumor field effect signature from MsigDB's C2 collection ($C' = 0.51$). SAT1, a gene in this signature, is up-regulated in the disease cohort, suggesting a metabolic switch in the monocyte-like cells from patients compared to the healthy controls

signatures distinguishing granulocytes and erythrocytes (Lian neutrophil granule constituents ($C' = 0.7$, FDR$<4.5 \times 10^{-3}$) and Hallmark heme metabolism ($C' = 0.72$, FDR$<4.5 \times 10^{-3}$), respectively) highlighted the granular neutrophil and erythrocytic arms of the trajectory. Furthermore, high values for a signature describing hematopoietic stem cell and progenitor populations (Ivanova hematopoiesis stem cell and progenitor, $C' = 0.4$, FDR$<0.04$) significantly localize to the undifferentiated core of the trajectory (Fig. 4c).

Notably, additional interesting signatures were found to be significant, emphasizing more nuanced biological processes occurring during hematopoiesis. For example, the granulocytic arm showed high signature scores for the *CBFA2T3* Targets signature ($C' = 0.89$, FDR$<4.5 \times 10^{-3}$, Fig. 4b). This signature includes genes that are up-regulated after *Mgt16* (or *CBFA2T3*) knockdown (i.e. those repressed normally by *Mgt16*), skewing HPCs to a granulocytic lineage and thus highlighting *Mgt16* as a key regulator of HPC lineage commitment. Additionally, the *KLF1* Targets ($C' = 0.83$, FDR$<4.5 \times 10^{-3}$, Fig. 4d) signature, which includes genes that are potential *EKLF* targets responsible for the failure of erythropoiesis, illustrates clearly that *KLF1* is an important regulator of the erythrocytic lineage. Taken together, these results show that the combination of signatures and latent trajectory models can emphasize important regulators of dynamic processes such as development.

As a complementary approach, *Vision* can perform analysis on numerical meta-data, such as QC measures. We make use of this function to identify numerical signatures which best distinguish different arms of the hematopoietic trajectory. Firstly, we observed that the ratio of genes detected in a cell (referred to as the cell detection ratio, or CDR) and the number of UMIs

had significant local autocorrelation as determined by the Geary $C$ statistic ($C' = 0.93$ and $C' = 0.90$, respectively, FDR$<4.5 \times 10^{-3}$ for both). Then, leveraging the label-based differential signature test, we find that these two meta-data variables are both associated with higher values in granulocytic cells (FDR$<4.5 \times 10^{-3}$, both, Supplementary Fig. 4b). Here, this may reflect the fact that granulocytes tend to have diameters twice the size of other white blood cells and erythrocytes. However, in other experiments where such a difference is not expected, such a result may signal that the data is confounded by technical noise and requires further correction through various forms of scaling or normalization[14,29].

## Discussion

Here we have presented *Vision*—a tool for exploration and functional interpretation of cell-to-cell similarity maps. *Vision* builds upon our previous work, FastProject[19], which was designed to annotate two dimensional representations of single cell data. Overall, the work described here provides a substantial increase in functionality, such as support for higher-dimensional latent spaces, explicit interpretation of pre-computed clusters, cellular trajectories, and other meta-data, as well as improved scalability. *Vision* also refines the core algorithms for signature analysis, most notably including a new autocorrelation score statistic which exhibits marked stability with respect to hyper-parameters (see the "Methods" section, Supplementary Fig. 1). Because *Vision* is able to operate on a variety of manifolds (latent spaces or trajectories from a broad array of methods) and scale to a large number of cells, it is well suited to enable the interpretation of single-cell RNA-seq data as modeling methods continue to evolve. Moreover, *Vision* offers greater flexibility to

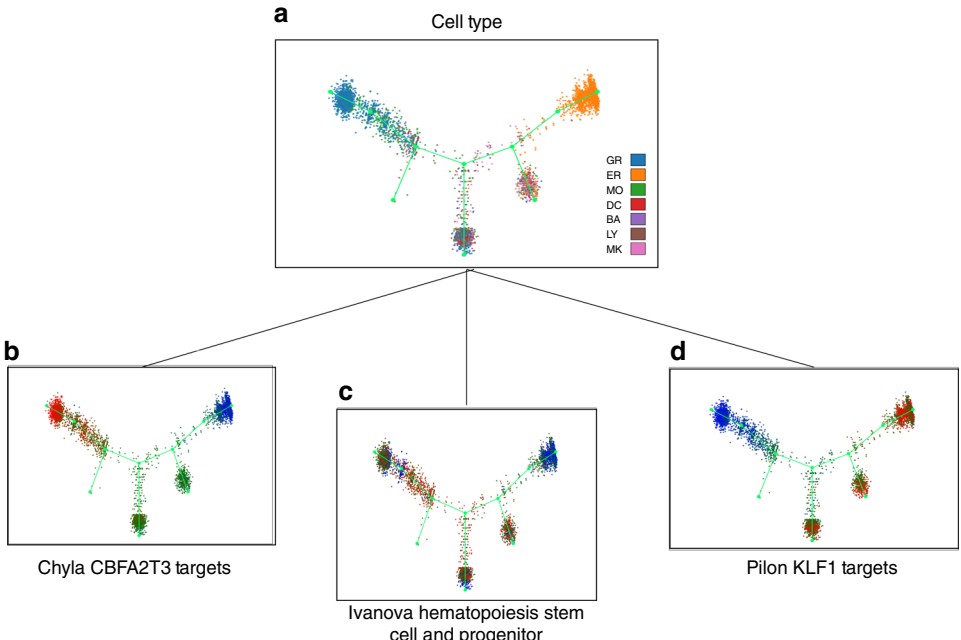

**Fig. 4** Annotating trajectories with *Vision* highlights transcriptional programs during hematopoiesis. Signatures can be used to describe functional variation along a trajectory, such as those computed with Monocle[7]. Here we have analyzed a set of 5432 hematopoietic progenitor cells (HPCs) isolated from bone marrow of adult mice. **a** The trajectory stratifies cells across seven major cell types identified in the original study: GR granulocytic, ER erythrocytic, MO monocytic, DC dendritic cell, BA basophilic or mast cell, LY lymphocytic, MK megakaryocytic. High-scoring transcriptional signatures can be used to annotate trajectory branches futher and highlight transcription factors involved in lineage commitment, such as **d** *Klf1* guiding the erythocytic trajectory and **b** *CBFA2T3* controlling the granulocytic trajectory from **c** an undifferentiated core characterized by the Ivanova hematopoiesis stem cell and progenitor signature

researchers because it can fit downstream of other methods, thus decoupling the choice of modeling algorithm and the subsequent interpretation.

A distinguishing feature of *Vision* is that it searches specifically for biological signals that are reflected in the latent cell manifold. While signals may exist that are not reflected by proximity in the manifold, the space of options there is much larger, resulting in a more difficult problem both in terms of identifying true signals and rejecting false ones. The approach used here is not only sensitive to large, global signals which stratify cells into groups, but also to more localized within-group variation, such as the interferon-beta signature which further stratifies both the naive and memory T cells of Fig. 2.

Signatures for analysis can be obtained from online databases, such as MSigDB[26] or users can define their own (based on, for example, the results of a differential expression test on a related experiment). Since signatures are often based on different experimental modalities (bulk RNA-seq or microarray), we have decided to use a simpler model where the gene-coefficients are restricted to ± 1. This also allows signatures of a similar size and positive/negative ratio to share a background distribution, which greatly reduces the computational requirement of empirical *p*-value estimation. Though a signature's gene-coefficients are of the same magnitude, individual genes will ultimately differ in their agreement with the aggregate signature expression. To assess this, the covariance between each gene's expression values and the signature score is computed and made available in the output report. Alternately, users can include signature scores computed in a different manner (such as that of ref. [37]) by adding the values as cell-level meta data. Signatures supplied in this way are evaluated using the same autocorrelation score as in the standard *Vision* analysis, while their significance is instead assessed using a per-signature permutation test (see the "Methods" section).

The results of a *Vision* analysis can be explored through the use of an interactive web-based graphical user interface (Fig. 1b and Supplementary Fig. 5). In this interface, top-scoring signatures are listed and can be visualized on the single-cell manifold through the use of two-dimensional projections, such as those provided by the user or generated internally with t-distributed stochastic neighborhood embedding[42]). If the latent space is a trajectory instead, the tree structure is visualized along with individual cells using various algorithms for tree embedding in two dimensions (e.g. refs. [43,][44]). To facilitate the interpretation of the data, *Vision*'s output report offers three possible views: a "signature-centric" view which highlights gene signatures deemed important via our analysis (label free and label-based); a "cell-centric" view, which allows users to independently analyze a subset of cells and view its properties (signature scores, expression of independent genes, meta-data) in comparison to the remainder of cells; a "gene-centric" view which allows users to view a single gene at a time. These three views enable a deeper dive into the results of three types of analysis, performed by *Vision*: a standard manifold mode (as in Figs. 2 and 3), a trajectory mode (as in Fig. 4), and an additional mode (dubbed *LCAnnotator*), which looks for correlation between individual components of the latent space (provided by the user or computed by *Vision* using PCA) and cellular properties (gene signatures, user-provided stratifications, and meta-data). See Supplementary Fig. 5 for the position of these and other features in the user interface.

*Vision* can accommodate 50,000 cells in about 30 min; however, to scale well into the hundreds of thousands of cells, *Vision* utilizes a micro-pooling algorithm in which the expression profiles of nearby cells in the similarity map are merged into representative "micro-clusters" (Supplementary Fig. 6a and the "Methods" section; note that micropooling was not necessary for the case studies described in this paper). Overall, we find that the

ordering of signatures in terms of their local autocorrelation remains consistent after micro-pooling and that the micro-clusters produced are biologically coherent (Supplementary Fig. 6b, c). Taken together, we find this approach heavily reduces the computation time—allowing the analysis of 500,000 cells with 20 cells per micro-cluster in under an hour while producing results consistent with the non-pooled analysis.

In comparison with other single-cell RNA-seq software tools, *Vision* augments the functionality of toolkits like Seurat[45], Scanpy[46], and scVI[11], which can be used for normalization, stratification, and labeling of the data prior to *Vision* analysis. *Vision* is distinct from visualization tools such as SPRING[47] in that it offers functional interpretation of the single-cell data. Moreover, *Vision* goes beyond standard workflows that provide gene set enrichment analysis on genes differentially expressed between groups (such as that offered by MAST[48]), by providing functional interpretation for inferred trajectories and for cases when the cells cannot clearly be clustered into groups.

Finally, in comparison to tools that can annotate important axes of biological variation without the need for a priori stratification (such as PAGODA[49], f-scLVM[50], and ROMA[51]), we demonstrate that *Vision*'s signature scores more effectively capture the underlying biological differences between samples and more precisely highlight crucial variation and sub-clusters in the data (Supplementary Fig. 7, Supplementary Fig. 8, and the section "Methods"). A detailed comparison of the analysis features available in *Vision* versus those available in other tools is provided in Supplementary Table 1.

In summary, *Vision* offers a scalable, automated, and easy-to-use tool for characterizing variation and heterogeneity in single cell RNA-seq data. As the number of methods for generating and then processing single cell measurements (e.g. CITE-seq for simultaneous protein and gene measurements[23]) increases, we anticipate that methods like *Vision*, which are able to integrate data at different levels and flexibly interpret the results of many pipelines will be in high demand. Finally, because the results of *Vision* can be made available through an interactive web-based report, we expect that it can be used to accelerate collaborations and further enable better reproducibility and communication of results from scRNA-seq studies.

## Methods

**Signature score calculation**. A signature is defined as a set of genes associated with some biological function or measured perturbation. Signatures may be *signed* in which there are two sets of genes, a positive set $G_{pos}$ and a negative set $G_{neg}$. Such a signature is used when describing an experimental perturbation or a comparison between two cell states in which some genes increase in expression while others decrease. Alternately, a signature may be *unsigned* in which case $G_{neg}$ is empty.

For each signature, a representative score is computed for every cell. This is calculated as the sum of expression values for positive genes minus the sum of expression values for the negative genes. For example, for signature $s$ and cell $j$ the score is computed as:

$$s_j = \frac{\sum_{g \in G_{pos}} e_{gj} - \sum_{g' \in G_{neg}} e_{g'j}}{|G_{pos}| + |G_{neg}|}$$

In the expression measure above, $e_{gj}$ is taken to be the normalized, log-scaled expression (e.g. log of counts-per-million + 1 or log of counts-per-thousand + 1) of gene $g$ in cell $j$. However, we have observed that even after performing standard normalization procedures on the expression values (e.g., regressing out technical covariates), signatures scores as defined above may still tend to be correlated with sample-level metrics (such as the number of UMIs per cell). To account for this, we z-normalize the signature scores using the expected mean and variance of a *random* signature with the same number of positive/negative genes. Specifically, for a signature score, $R_j$ in cell $j$ derived from a random signature with $n$ positive genes and $m$ negative genes:

$$E[R_j] = \frac{n - m}{n + m} \bar{e}_j \qquad var(R_j) = \frac{\sigma_j^2}{n + m}$$

where $\bar{e}_j$ and $\sigma_j^2$ represent the mean and variance, respectively, of the expression

values for cell $j$. The final, corrected signature score, $s'_j$ is computed as

$$s'_j = \frac{s_j - E[R_j]}{var(R_j)^{\frac{1}{2}}}$$

**Local autocorrelation calculation**. To compute the extent to which a signature can explain the variation in a cell-to-cell similarity map, we make use of the Geary's $C$ statistic for local autocorrelation. This statistic is defined as

$$C = \frac{(N-1) \sum_i \sum_j w_{ij}(x_i - x_j)^2}{2W \sum_i (x_i - \bar{x})^2}$$

where $w_{ij}$ represents the weight between cells $i$ and $j$ in some similarity map, $x_i$ is a value of interest, $N$ is the total number of cells, and $W$ is the sum of all weights. In our case, the value of interest (i.e. $x$) are the ranks of the normalized signature score in each cell. The weights, $w_{ij}$ between cells $i$ and $j$ are set to be nonzero and positive for cells nearby in the provided latent space (details follow).

In this way, the Geary's $C$ provides a measure of how similar the signature ranks are for neighboring cells given a latent mapping. For the interactive output report, we report $C' = 1 - C$ as the autocorrelation effect size so that a 0 intuitively represents no autocorrelation and a 1 represents maximal autocorrelation.

To compute the cell–cell weights, $w_{ij}$, for the Geary's $C$, first the $k = N^{\frac{1}{2}}$ nearest neighbors are evaluated for each cell in the provided latent map. If the input map is a latent space, this is evaluated using euclidean distance in the latent space. If the input map is a tree or trajectory, this is evaluated using the path distance along the trajectory. To accommodate the variety of latent trajectory methods which have been developed for single-cell RNA-seq, we make use of the Dynverse package[12] which rectifies the output of over 50 such methods to a common trajectory model. While by default $k$ scales as the square-root of the number of cells, this value can be set directly by the user (though we have found the results to be relatively insensitive to the neighborhood size, Supplementary Fig. 1).

Once distances and neighbors have been determined, cell–cell weights can be calculated. For cells which are $k$-nearest neighbors, the weight is evaluated as:

$$w_{ij} = \exp(-d_{ij}^2 / \sigma_i^2)$$

where $d_{ij}$ is the distance between cell $i$ and $j$ in the latent mapping and $\sigma_i^2$ is the distance to the kth-nearest neighbor of cell $i$. If cell $j$ is not a $k$-nearest neighbor of cell $i$, then $w_{ij}$ is taken to be 0.

**Assessing significance of autocorrelation scores**. To evaluate the significance of the autocorrelation scores for each signature, a set of random signatures are generated (genes drawn from the set of genes in the input expression matrix), and autocorrelation scores on these signatures are computed to act as an empirical background distribution. The p-value for signatures is then computed as $p = \frac{x+1}{n+1}$ where $x$ is the number of background signatures with a higher autocorrelation score and $n$ is the total number of background signatures. These p-values are then corrected for multiple-testing using the Benjamini–Hochberg procedure.

To avoid the computational cost of generating a random background for every evaluated signature, we instead create five background signature groups which encompass the range of signature sizes (number of genes) and balance (ratio of pos/neg genes) in the input signature set. This is sufficient as we have observed that the distributions of random signature p-values are very similar even when the size and background are not perfectly matched. The size and balance of background signature groups is evaluated by clustering all input signatures by their $\log_{10}(\text{size})$ and balance using k-means with $k = 5$. Cluster centers are then used for the background group sizes and balance, and cluster assignments are used to match signatures under test with the closest background.

**Micro-pooling**. *Vision* employs a micro-pooling algorithm to partition the input expression matrix and pool cells together, resulting in a reduction of computational burden for data sets consisting of a large number of cells. The algorithm begins by applying gene filters to the input expression matrix: genes are first thrown out that are not expressed in at least 10% of cells and then highly variable genes are selected as in ref. [19]. Next, we project the filtered matrix down to 20 dimensions using PCA. Then for the $N$ cells in the expression matrix, the KNN graph (with $K = \text{sqrt}(N)$) is computed in this reduced space.

Initially, this KNN graph is clustered using the Louvain algorithm, an efficient community detection algorithm. These clusters are further partitioned with K-means until each cluster consists of at most $P$ cells per partition ($P$ can be controlled by the user).

"Micro-clusters" are then generated using these partitions. For each partition, we create a micro-cluster whose gene expression values are defined as the average gene expression values for each cell in the partition. More specifically, for gene $i$ in micro-cluster $z$ generated from partition $P_z$, the expression value for this gene $e_{iz}$ is equal to:

$$e_{iz} = \frac{1}{|P_z|} \sum_{j \in P_z} e_{ij}$$

Finally, an expression matrix consisting of these micro-clusters of dimension $\mathcal{O}(N/P) \times G$ is returned and used for downstream analysis.

**Assessing biological coherence for micro-clusters**. We assessed the biological coherence of the micro-clusters with a dataset consisting of simultaneous epitope and gene expression profiles of single cells, published in ref. [23] (Gene Expression Omnibus accession GSE100866). For $\sim$ 9000 cord blood mononuclear cells (CBMCs), we performed micro-pooling on the *transcriptional* data to create micro-clusters with at most 20 cells per cluster with the gene expression data. Then we analyzed the relative variation in *protein* within each micro-cluster. More specifically, we reported the ratio of intra-micro-cluster standard deviations to the overall standard deviation of the protein across the entire population of cells:

$$\sigma_{pi}^2 = \frac{1}{|S_i|} \sum_{j \in S_i} (x_{pj} - \overline{x}_{pi})^2$$
$$\sigma_p^2 = \frac{1}{N} \sum_k^N (x_{pk} - \overline{x}_p)^2$$
$$r_{pi} = \frac{\sigma_{pi}}{\sigma_p}$$

where $\sigma_{pi}^2$ is the variance across the cells in micro-cluster $i$ for protein $p$, $\sigma_p^2$ is the population-wide variance of protein $p$, $x_{pj}$ is the abundance of protein $p$ in cell $j$, $\overline{x}_{pi}$ is the mean abundance of protein $p$ across cells in micro-cluster $i$, $\overline{x}_p$ is the mean abundance of protein $p$ across the population, and $r_{pi}$ is the ratio between the standard deviations for a particular micro-cluster $i$. We then report the distribution of ratios across all micro-clusters for each protein separately, as presented in Supplementary Fig. 6b.

**Cell–cell similarities from trajectories**. *Vision* interfaces with the Dynverse package[12] to process trajectories in an analysis pipeline. The results of running a trajectory with Dynverse is an abstracted trajectory model which *Vision* is able to ingest and process. Most essential to the *Vision* pipeline are two components of the Dynverse model: (a) the "milestone" network detailing the topology of the trajectory (e.g., in a developmental process, milestones would be important cell states or types and the topology would represent how these states are related to one another) and (b) the progress of cells along this network (i.e., where cells lie between the important milestones).

Using the milestone network and the progress of cells between each pair of milestones (i.e. a "pseudotime") we define cell–cell similarities according to the tree-based geodesic distances. Given this cell–cell similarity map, we can then perform the same autocorrelation score evaluation for all signatures as described above.

*Vision* visualizes the trajectory by first applying a method to visualize the milestone network and then projecting the cells onto their assigned edges, where their locations between edges are proportional to their pseudotime. *Vision* uses a variety of methods for visualizing the milestone network such as Fructerman–Reingold[44] and Davidson–Harel[43]. Importantly, to help visualize edges where many cells are located, a small amount of jitter to each cell's position is added perpendicular to its assigned edge.

**Differential signature analysis**. Similar to a differential gene expression test, *Vision* performs a test to identify which signatures' scores are differential among a particular group of cells. These groups of cells are defined using any input meta-data of a categorical nature (i.e. discrete variables such as disease status or clustering assignments). For each supplied categorization, we test for signatures that are differential, by performing a Wilcoxon rank-sum test for every 1 vs. All comparison. For these tests, the effect size is reported by converting the rank-sum test statistic to its equivalent area under the ROC curve (AUROC) along with the associated p-value. The results of these tests represent one of the "label-based" analyses performed by *Vision* and are available for browsing in the output *Vision* report.

**Autocorrelation score of discrete meta-data**. The Geary's C cannot be used to evaluate the autocorrelation of discrete meta-data variables (such as donor or batch), and so instead, *Vision* uses a procedure based on the chi-squared test. First the local distribution of the variable is computed around each cell. Then, these local distributions are aggregated into a square contingency table whose rows represent the distribution of the variable as observed local to the cells of each value. For example, if run on a batch variable, the row representing batch $x$ will contain proportions of each batch as estimated from the local neighborhoods of cells in batch $x$. This table is then evaluated with the chi-square test.

More concretely, first, for each cell, $i$, a local proportion for each variable value $m$ is evaluated as

$$\hat{c}_{im} = \sum_j w_{ij} \mathbf{I}_m(c_j)$$

Here, the weights $w_{ij}$ are computed from the manifold using the same procedure described above for transcriptional signatures, $c_j$ represents the value of the discrete variable of interest in cell $j$, and $\mathbf{I}_m(x)$ is an indicator function that takes on a value of 1 if $x = m$ and 0 otherwise. From these values, the contingency table $\mathbf{X}$ is computed as

$$X_{lm} = \sum_i \hat{c}_{im} \mathbf{I}_l(c_i)$$

The chi-squared test is then performed on this contingency table $\mathbf{X}$ to estimate a p-value.

Because of the large number of cells involved in modern scRNA-seq experiments, it is possible to achieve a significant p-value for an autocorrelation effect that is too weak to be of interest. Accordingly, *Vision* also reports the effect size as the Cramer's V in addition to the p-value. This indication of effect size ranges from 0 (no autocorrelation) to 1 (perfect autocorrelation), and provides an additional means to rank and categorize potentially confounding effects.

**Analysis of single-cell expression profiles from lupus cohort**. Gene expression counts for 29,066 PBMCs across eight donors from ref. [32] were downloaded from NCBI GEO (accession GSE96583), as well as annotations for stimulated/unstimulated, cell type, and single/doublet. Cells marked as 'doublet' were filtered prior to downstream analysis. When subsetting by cell type, the cell type labels from the original study were used. Prior to input into *Vision*, gene expression data was filtered to remove genes detected in <10 cells and normalized by scaling the sum of UMIs in each cell to the median number of UMIs across all cells. When isolating the stimulated and unstimulated CD4+ cells individually, cells belonging to cluster 3 (annotation from original study) were excluded as these appeared to be proliferating T cells whose large difference from the rest of the cells served to mask more fine-grained heterogeneity. On the full set of PBMCs, signatures for analysis were taken from MSigDB[20] (Hallmark and KEGG collections)[33], and CiberSort[52]. For the CD4+ T cells, an additional 707 signatures which were derived from perturbations involving CD4+ T cells were added from the MSigDB C7 collection. Two-dimensional visualizations of cells were computed by first taking the top 30 principal components and then reducing further with the tSNE algorithm using a perplexity of 30.

**Analysis of single-cell expression profiles of AML cohort**. Raw gene expression counts for 38,410 cells from 40 bone marrow aspirates, including 16 AML patients and five healthy donors, were downloaded from NCBI GEO, accession GSE116256. In addition to the counts, we also obtained meta-data pertaining to each of the cells including the patient donor, predicted cell type, and any mutations observed in the cell. Before analyzing with *Vision*, we performed batch correction with scVI[11] using patient ID as the batch variable, filtered out genes with the Fano-based filter implemented in *Vision*, and scaled the sum of UMIs in each cell to the median number of UMIs across all cells. In the monocyte-only analysis, we subset the cells to include only those labled as "Monoctye", "Pro-Monocyte", "Monocyte-like", and "Pro-Monocyte-like", leaving 7780 cells. Gene filtering and UMI scaling were done separately for this subset of the data. Two-dimensional visualizations of the cells in both analyses were computed by first taking the 10 dimensional latent space found with scVI and then reducing further with tSNE, perplexity 30.

**Analysis of hematopoietic stem cells (HSCs)**. The expression profiles of 5432 HSCs were obtained from NCBI GEO, accession GSE89754; in this analysis, we used the raw UMI counts of the basal bone marrow HSCs (specifically, GSM2388072)[41]. Before computing the trajectory, we first filtered the genes using the gene set that the original authors used, and removed cells which the authors flagged as not passing their own internal filters. Monocle2[7] was used to infer the developmental trajectory (using the "log" normalization scheme), and we then wrapped the final inferred trajectory with dynverse[12]. The cell types reported here are those used in the original study.

For signature score evaluation we then scaled the raw number of UMIs per cell to the median UMI count across the dataset. A *Vision* object was created with these scaled counts, signatures consisting of both the Hallmark and C2 MSigDB collections[20], and the Dynverse-wrapped Monocle trajectory.

**Comparison of *Vision* to existing tools**. To broadly exhibit the unique features of *Vision*, we conducted a qualitative comparison of *Vision* to other similar tools that seek to combine functional analysis and visualization for large scRNA-seq datasets. Supplementary Table 1 summarizes this comparison for a panel of methods, including Spring[47], CCS[53], ROMA[51], PAGODA[49], MAST[48], Scanpy[46], and Seurat[45]. As demonstrated, *Vision* has a comprehensive set of analysis capabilities, some of which (e.g., annotating trajectories or adding meta data to the analysis) are unique whereas others (e.g., performing cluster-based, but not cluster-free analysis) are only partially present in other packages.

A key distinguishing feature of *Vision* is its ability to provide biological annotations for cell to cell variability in both a cluster-based and cluster-free manner. While most existing tools are restricted to the former type of analysis, it is often the case in single-cell datasets that cells do not neatly partition into groups. For these instances, the variation within a group is of primary interest (see example in Fig. 2). Of the reference tools surveyed above, three methods—ROMA[51], PAGODA[49], and f-scLVM[50]—were designed to identify and annotate important

axes of biological variation in a dataset without the need for a priori stratification of the cells.

To demonstrate the value of *Vision* compared with these methods, we ran a number of evaluations. We first used the ROMA method on the interferon beta-stimulated CD4 T cell cluster from ref. [32] with the Hallmark (MSigDB[20]) signature set, as well as signatures taken from ref. [33]. ROMA did not select any of the signatures as significant despite the clear coordinated variation exhibited among the cells for the IFNb response signature (from ref. [33]) and the similar Interferon Alpha/Gamma response signatures from the Hallmark library. We then tested the PAGODA method on the same set of cells, using the full set of signatures we had previously run with *Vision*. First, we examined the effectiveness of the per-cell signature scores reported by each tool. It is expected that CD4 T cells will partition according to naive/memory axis of variation and this appears to be true in this data as well based on *CCR7* and *S100A4* expression (Supplementary Fig. 7a). We clustered the cells into these two groups using an unsupervised algorithm (Louvain with resolution 0.3 and 30 neighbors) and compared the scores derived from three CD4 T cell naive vs. memory signatures found in the MSigDB database (Supplementary Fig. 7b). It is observed that the scores produced by *Vision* better distinguish the two clusters while those produced by PAGODA show almost no difference between the two groups for 2/3 of the signatures.

In addition to signature scores, each tool also provides a test statistic which can be used to rank signatures according to their overall affect on variation within the samples ($C'$ local autocorrelation for *Vision* and adjusted-z overdispersion for PAGODA). To compare the effectiveness of these statistics, we examined how they change between the stimulated and unstimulated CD4 cell subsets (Supplementary Fig. 7c). It can be seen that most signatures lie roughly on the diagonal, which is reflective of the fact that both subsets should have common biological variation (e.g., naive vs. memory, effector vs. regulatory, etc.). However, *Vision* greatly emphasizes a group of signatures which have uniquely high $C'$ values in the stimulated cells only (Supplementary Fig. 2a), which consists of the IFNb stimulation signature from ref. [33], as well as the Interferon Response signatures form the MSigDB Hallmark collection. When examining the overdispersion results from PAGODA, however, this change is much less pronounced (Supplementary Fig. 7c, second panel), and could easily be mistaken for normal variation between the two samples.

Lastly, we sought to compare *Vision* with f-scLVM[50]; however, we believe that f-scLVM is primarily designed to solve a different problem than *Vision*. f-scLVM aims to decompose cellular variation using a set of supervised (signature-based) and unsupervised factors, finding a minimal set of factors which can be used to describe the data. *Vision* instead aims to describe an existing cell–cell similarity map through the use of signatures. The distinction is that when multiple factors correlate, f-scLVM will attempt to select a single factor, either down-weighting the contribution of alternative factors or selecting their weights so that they fit a different component of cellular variation (Supplementary Fig. 8a, b). The results for individual factors will be highly dependent on the presence of other, correlated factors. *Vision*, on the other hand, evaluates the signature scores and signature autocorrelation of each factor in an independent manner. As a consequence of this, f-scLVM may be better suited for cases when a small set of candidate uncorrelated gene signatures can be selected in advance, while *Vision* is more suited to exploratory analyses in which samples are evaluated against a large library of signatures. As described above with PAGODA, we further compared the f-scLVM importance scores between the stimulated and unstimulated CD4 T cells from ref. [32] (Supplementary Fig. 7c). In this instance, a smaller library of signatures (only the MSigDB Hallmark collection) was used due to large runtimes when using a larger set of signatures (Supplementary Fig. 8c). Here, though f-scLVM does assign a higher importance score to the interferon gamma response signature within the stimulated cells, the distinction between the two samples is much less pronounced than what is shown by *Vision* as f-scLVM ranks both interferon response signatures among the top few contributing factors for both samples. Additionally, the approach used in *Vision* scales much better with the number of signatures under consideration (Supplementary Fig. 8c). Finally, while f-scLVM constructs a latent model as part of its method, *Vision*'s construction is more flexible in that it can operate on the results of any latent model of cell–cell variation.

**Stability analysis of $K$, the number of neighbors**. We performed two tests for the number of neighbors used in the signature autocorrelation test:

The first test consisted of performing standard autocorrelation analysis on a subpopulation of 7780 monocyte cells from the larger set of AML cells. This analysis used default settings for the number of neighbors, $K = \lceil sqrt(N) \rceil = 89$. We then kept the set of signatures whose $q$-value was less than or equal to 0.05 —this set is denoted as $S$. Then for values of $K' \in \{1, 10, 20, 30, 40, 50, 60, 7080, 90, 100, 120, 150, 200\}$, we calculated the consistencies of these signatures in $S$. For each of these analyses, we computed the rank correlation between consistencies of signatures in $S$ found with $K$ and some $K'$.

The second analysis was performed on the same subpopulation of cells. We began the same way in evaluating local autocorrelation with $K = \lceil sqrt(N) \rceil = 89$ and finding the set of significant signatures $S$ with a $q$-value of <0.05. Then for each $K' \in \{1, 10, 20, 30, 40, 50, 60, 7080, 90, 100, 120, 150, 200\}$ we found the set of signatures significant in the anlaysis, $S'$. Then for each pair of $S$ and $S'$, we

computed the Jaccard Index to assess how the sets of significant signatures changed between runs with different numbers of neighbors.

**Reporting Summary**. Further information on research design is available in the Nature Research Reporting Summary linked to this article.

## Data availability

This data used in this manuscript has been previously published and is available in the NCBI GEO repository at accessions GSE100866 (interferon-stimulated PBMCs[32]), GSM2388072 (AML[36]), and GSE96583 (Hematopoietic Stem Cells[41]).

## Code availability

The *Vision* software is available on GitHub (http://www.github.com/YosefLab/VISION) and archived at https://doi.org/10.5281/zenodo.3345985.

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

## Acknowledgements

The authors would like to thank the members of the Yosef and Ye labs for their helpful discussions in the development of this project. This work was funded by NIH-NIAID grant U19 AI090023 (N.Y., D.T.), NIH Training Grant T32 GM067547 (M.J.), and Chan-Zuckerberg Initiative 2018-18034. N.Y. is also a member of the Chan Zuckerberg Biohub investigator program.

## Author contributions

D.D., M.J. and N.Y. conceived of core algorithms and wrote the manuscript. M.J. and D.D. designed the software package. T.A. conceived of and implemented algorithms supporting the use of inferred trajectories. M.S. and C.J.Y. advised on the interpretation of results involving the Lupus cohort.

## Additional information

**Competing interests:** The authors declare no competing interests.

