## [Peer Review File · Nature Communications]

Reviewers' comments:

Reviewer #1 (Remarks to the Author):

This paper develops a tool called VISION that is designed to aid interpretation of intercellular heterogeneity using annotated gene sets. I think that this is an important problem and that this paper makes some novel contributions. However, I have some questions about the limitations of the approach and relationship to previous work. I also have some suggestions that may make the method more useful and increase the novelty.

Major Comments:

1. How much of an advantage does this approach provide compared to simply clustering and finding marker genes? In the main examples shown in the paper, it appears that the sources of variation identified using the autocorrelation statistic could have been more easily detected by clustering (perhaps at higher resolution or in a second round of sub-clustering) and then finding differentially expressed genes or performing gene set enrichment analysis. For example, the naïve vs. memory B cell split seems like it should be readily detectable through clustering. I think there probably are cases where the autocorrelation is better, but this is a key point that needs to be made clearly to convince potential users of the tool why this approach is better.
2. A fundamental assumption of this approach is that the sources of variation are “localized” among similar cells in the knn graph. How will VISION handle sources of variation that are not localized in this sense, but perhaps are orthogonal to or at odds with the dominant biological signals that define the knn graph? For example, neurons of different types may show activation of the immediate early gene pathway. Is there any way to extend VISION to discover such “non-localized” variation? In some sense, discovering signals of this sort may be more useful, because by definition they cannot be discovered through clustering and they define uncharacterized axes of variation.
3. Is there any way to extend the approach to discover unannotated sources of variation? One hope of scRNA-seq experiments is that we can update annotations and discover new gene sets.
4. How does VISION relate to factor analysis models, which can discover both annotated and unannotated sources of variation? For example, f-sLVM can be used to perform a similar functional interpretation by identifying the “relevance” of known signatures and unknown but strongly correlated gene sets (<https://genomebiology.biomedcentral.com/articles/10.1186/s13059-017-1334-8>). This seems like an important class of methods that should be cited and discussed.
5. VISION lacks individual gene weights for each signature (equivalent to implicitly assuming that each gene contributes equally to the signature). Is there a way to remove this limitation? Can you assess by simulation how robust your results are in the presence of deviations from equal gene contribution? In other words, do you lose detection sensitivity if only a few genes within a signature have large weights in the real data, but you assume that all genes contribute equally?
6. The choice of k as \sqrt{N} seems quite arbitrary. You should include an analysis in which you vary k and assess how this changes the results.

Minor Comments:

1. Is the name VISION an acronym? If so, it should be defined. If not, perhaps it should not be written in all capital letters.
2. The sentence at the end of the first paragraph seems like an oversimplification. Perhaps the point you want to emphasize is that, although case-control studies are still important with scRNA-seq, investigation of the heterogeneity within a single population of cells is now possible.
3. Much of the text from the screenshot in Fig. 1B is illegible. Also, the legend for panel B should explain the outputs more clearly. What dataset is this, and what does each piece of data indicate?
4. What do the colors in Fig. 2A represent? I assume that this is the inferred expression for each signature but it is not stated in the caption.
5. For Fig. 2 and Fig. S3, it would be helpful to show, perhaps with a bar plot, the top-scoring signatures and their respective scores. The current presentation of the data does not effectively show the process by which the signatures shown here were identified.
6. Fig. S6B: the signature labels are too small to read.

Reviewer #2 (Remarks to the Author):

In the manuscript entitled "Functional Interpretation of Single-Cell Similarity Maps", the authors present their software package VISION for automated annotation and visualization of single cell RNA-Sequencing data. Other reviewers will address the technical aspects of VISION compared to other software packages in this space. My review is instead on the use and interpretation of single cell data.

The authors make use of several existing, well characterized datasets in their paper which affords one the ability to benchmark their software compared to other packages. And there clearly are features of their software that are informative and may perform better in some ways from existing packages. As users of these tools know, the pace of evolution of software for analysis of single cell RNA-Sequencing datasets is very rapid.

While the use of published data was appropriate and appropriately attributed the primary source studies, I find it very odd how the bulk of the SLE data used in this paper was from a "subset of a much larger, unpublished dataset" that will soon be made public as stated in the authors disclosures. Description of this cohort was sparse in the methods and missing key information traditionally expected when data from a human subject research project is published. It is unclear if any of the authors on this manuscript are a key member of the parent study, or if they are direct collaborators. The reason the use of this data is problematic, is that at no point in this manuscript are we provided the details of the recruitment of these subjects, who the primary authors of those studies are, whether IRB approved study, details of the demographics of those subjects, or details of the clinical characterization of those subjects. Yet the authors use these data in no less than five figures in the manuscript and a large part of the discussion is based on these figures based on analysis of this SLE cohort. In the analysis, the authors point out, rightly so, that the donor-donor variation far exceeds the cell-based expression variation in these data. With 12 SLE patients and 4 controls in this cohort, that would be totally expected. We don't know if these were age, gender or race matched cases and controls. We don't know anything about the ACR criteria of diagnosis or the range of SLEDAI disease activity scores. The interferon, TNF and neutrophil signatures they describe are well known, expected signatures that are seen in many cell types. However, the authors use these data on a cohort for which we do not fully know the details of the experimental design, to make broad generalizations about cell types and gene expression signatures in SLE which may be true or may not be true. If the study was fully reported, well controlled and designed, as we have to assume that the "much larger unpublished study" might be, then maybe these observations could be supported. It is unfortunate that the authors have not elected to await the publication of the larger parent study to allow these data on SLE to be properly reviewed and published. I would worry that premature publication of a potentially superficial analysis of these data would be a disservice to both the primary study investigators, the patients who provided samples to that study and to the SLE research community at large.

Reviewer 1 (responses below in blue)

This paper develops a tool called VISION that is designed to aid interpretation of intercellular heterogeneity using annotated gene sets. I think that this is an important problem and that this paper makes some novel contributions. However, I have some questions about the limitations of the approach and relationship to previous work. I also have some suggestions that may make the method more useful and increase the novelty.

Major Comments

1. How much of an advantage does this approach provide compared to simply clustering and finding marker genes? In the main examples shown in the paper, it appears that the sources of variation identified using the autocorrelation statistic could have been more easily detected by clustering (perhaps at higher resolution or in a second round of sub-clustering) and then finding differentially expressed genes or performing gene set enrichment analysis. For example, the naïve vs. memory B cell split seems like it should be readily detectable through clustering. I think there probably are cases where the autocorrelation is better, but this is a key point that needs to be made clearly to convince potential users of the tool why this approach is better.

We have found that single-cell data frequently results in more continuous phenotypes. For example, when attempting to discern between different T cell subsets, different regions of the CD4 cluster may exhibit characteristics of memory or effector or regulatory T cells without clear boundaries between them [1,2]. In this case, relying on a clustering algorithm first introduces more variability in the analysis in terms of the choice of algorithm and its hyperparameters. A second scenario where a cluster-free analysis is useful is shown in the reworked example with IFN β -stimulated CD4 T cells (See Figure 2). Here, there is variable activation of the interferon-stimulated genes that occurs within both the naïve and memory T cell clusters and this pattern of variation is identified by VISION. In this way, multiple signatures may induce different stratifications of the cells into groups or along gradients that may not be captured by a single clustering analysis. Here the Naïve vs. Memory signature stratifies the cells into two groups while the IFN β -stimulation signature stratifies the cells into two alternate groups. We now make this important subject more explicit (p. 5 and 9).

2. A fundamental assumption of this approach is that the sources of variation are “localized” among similar cells in the knn graph. How will VISION handle sources of variation that are not localized in this sense, but perhaps are orthogonal to or at odds with the dominant biological signals that define the knn graph? For example, neurons of different types may show activation of the immediate early gene pathway. Is there any way to extend VISION to discover such “non-localized” variation? In some sense, discovering signals of this sort may be more useful, because by definition they cannot be discovered through clustering and they define uncharacterized axes of variation.

The goal of this paper is to annotate a given manifold by associating different regions with function. However, the comment made by the reviewer is indeed valid as there can be signals that are not associated with localization in the low-dimensional representation in hand or not distinct enough to be captured in the manifold modeling procedure. The space of options in these regimes is much larger, and so the problem becomes more difficult - both in terms of identifying positive signals and rejecting noise-induced false positives.

However, to specifically address your concern with non-localized signals, these should still be captured by the analysis in *Vision*. For example, if cells are separated by cell-type first and by some more continuous phenotype second, you would expect a visualization of a signature score to show multiple clusters, each with an internal gradient. In this case, a cell's neighbors would still have similar scores (more so than by chance). This occurs in our analysis with the interferon-beta signature in the stimulated CD4 T cells (Figure 2C). Additionally, this should still hold as the effect becomes more non-localized, up until the point where the pattern of interest varies greatly within a cell's immediate neighborhood. Finally, depending on the user's input, the software can readily explore localized patterns in more minor axes of variation (e.g., focusing on PCs further along the eigenvalue spectrum and excluding the first ones), which will not necessarily dominate localized patterns in a standard analysis. We now relate to this point in the Discussion section (p. 9).

3. Is there any way to extend the approach to discover unannotated sources of variation? One hope of scRNA-seq experiments is that we can update annotations and discover new gene sets.

While we agree that this could be very useful, annotating sources of variation de-novo without the use of signatures is beyond the scope of our tool. However, *Vision* can still greatly facilitate such an analysis when combined with other tools and data sets. For instance, one can perform de-novo annotation analysis in a given data set of choice (e.g., using clustering and differential expression, or with topic modeling) and then inspect the effect of the resulting sources of variation in another data set. Notably, these sources of variation resulting from such "external" de-novo analysis can be formalized either as gene signatures (directed or not), or more generally as a "black box" function that given a transcriptional profile provides a quantitative score (e.g., cell state classifier[3]). In the former case, the results can be processed in *Vision* using its standard gene signature API. In the latter case, the quantitative scores can be computed outside *Vision* and then provided as quantitative meta-data (which will be subject to a similar type of localization analysis). We now discuss this important point in the discussion section (p. 9)

4. How does VISION relate to factor analysis models, which can discover both annotated and unannotated sources of variation? For example, f-scLVM can be used to perform a similar functional interpretation by identifying the "relevance" of known signatures and unknown but strongly correlated gene sets (<https://genomebiology.biomedcentral.com/articles/10.1186/s13059-017-1334-8>). This seems like an important class of methods that should be cited and discussed.

We agree with this recommendation and have added some discussion of and comparison with f-scLVM to the section on method comparisons in the supplement. We find that the method of f-scLVM is less suitable to the task of testing large libraries of signatures as each signature's per-cell scores and overall ranking are influenced by other (correlated) signatures. Additionally the computation time scales fairly poorly as the number of candidate signatures is increased (Figure S7).

5. VISION lacks individual gene weights for each signature (equivalent to implicitly assuming that each gene contributes equally to the signature). Is there a way to remove this limitation? Can you assess by simulation how robust your results are in the presence of deviations from equal gene contribution? In other words, do you lose detection sensitivity if only a few genes within a signature have large weights in the real data, but you assume that all genes contribute equally?

Though we can see how this enhancement would be useful in some cases, we have decided not to add it at this time as most signature databases currently do not support per-gene weights. Additionally, since some signatures may come from very different data modalities (bulk; possibly with microarrays, as in the immunological signatures of MSigDB), we believe it is generally better to keep the model simpler. Finally, the use of per-gene weights would necessitate creating an empirical background on a per-signature basis, resulting in a dramatically increased runtime for analysis.

As an alternative, though, users can compute signature scores separately (using whatever method; e.g., scANVI[3]) and input these into *Vision* as numerical cell-level meta-data. The local autocorrelation would be computed in the same manner, but for significance a permutation background is used instead. Additionally, in our user interface we now provide a ranking of genes within a signature based on their contribution (covariance) with the signature score. This is available in the output report so that user's can properly identify signatures primarily driven by few genes. We now discuss this important point in the discussion section (p. 9)

6. The choice of k as \sqrt{N} seems quite arbitrary. You should include an analysis in which you vary k and assess how this changes the results.

A stability analysis has now been added as a supplementary figure (S1) - we found the procedure to be very stable as K varies.

Minor Comments

1. Is the name VISION an acronym? If so, it should be defined. If not, perhaps it should not be written in all capital letters.

The name is not an acronym, and following this comment we have changed it to *Vision* in the revised manuscript to avoid confusion.

2. The sentence at the end of the first paragraph seems like an oversimplification. Perhaps the point you want to emphasize is that, although case-control studies are still important with scRNA-seq, investigation of the heterogeneity within a single population of cells is now possible.

We have edited this part of the introduction and removed this sentence (p. 1).

3. Much of the text from the screenshot in Fig. 1B is illegible. Also, the legend for panel B should explain the outputs more clearly. What dataset is this, and what does each piece of data indicate?

This figure serves to just show the output interface in a few different configurations, and we did not intend for every detail to be readable in these summary views. To provide a more detailed view, we have now added more supplemental figures to showcase the interface with larger screenshots where text is legible without zooming in (Figure S8).

In addition, we have several examples available on our Github (and below) demonstrating the output report on data used in this manuscript.

- Lupus Stimulated CD4 T cells (Kang, Subramaniam, and Targ et al., Nature Biotech 2017)
- AML Monocytes (Galen et al., Cell 2019)
- Hematopoiesis (Tusi et al., Nature 2018)
- Cite-seq PBMCs (Stoeckius et al., Nature Biotech 2017)

4. What do the colors in Fig. 2A represent? I assume that this is the inferred expression for each signature but it is not stated in the caption.

This figure has been replaced and colorbar legends have been added.

5. For Fig. 2 and Fig. S3, it would be helpful to show, perhaps with a bar plot, the top-scoring signatures and their respective scores. The current presentation of the data does not effectively show the process by which the signatures shown here were identified.

No longer relevant - this figure has been replaced

6. Fig. S6B: the signature labels are too small to read.

No longer relevant - this figure has been replaced

Reviewer 2 (responses below in blue)

In the manuscript entitled “Functional Interpretation of Single-Cell Similarity Maps”, the authors present their software package VISION for automated annotation and visualization of single cell RNA-Sequencing data. Other reviewers will address the technical aspects of VISION compared to other software packages in this space. My review is instead on the use and interpretation of single cell data.

The authors make use of several existing, well characterized datasets in their paper which affords one the ability to benchmark their software compared to other packages. And there clearly are features of their software that are informative and may perform better in some ways from existing packages. As users of these tools know, the pace of evolution of software for analysis of single cell RNA-Sequencing datasets is very rapid.

While the use of published data was appropriate and appropriately attributed the primary source studies, I find it very odd how the bulk of the SLE data used in this paper was from a “subset of a much larger, unpublished dataset” that will soon be made public as stated in the authors disclosures. Description of this cohort was sparse in the methods and missing key information traditionally expected when data from a human subject research project is published. It is unclear if any of the authors on this manuscript are a key member of the parent study, or if they are direct collaborators. The reason the use of this data is problematic, is that at no point in this manuscript are we provided the details of the recruitment of these subjects, who the primary authors of those studies are, whether IRB approved study, details of the demographics of those subjects, or details of the clinical characterization of those subjects. Yet the authors use these data in no less than five

figures in the manuscript and a large part of the discussion is based on these figures based on analysis of this SLE cohort. In the analysis, the authors point out, rightly so, that the donor-donor variation far exceeds the cell-based expression variation in these data. With 12 SLE patients and 4 controls in this cohort, that would be totally expected. We don't know if these were age, gender or race matched cases and controls. We don't know anything about the ACR criteria of diagnosis or the range of SLEDAI disease activity scores. The interferon, TNF and neutrophil signatures they describe are well known, expected signatures that are seen in many cell types. However, the authors use these data on a cohort for which we do not fully know the details of the experimental design, to make broad generalizations about cell types and gene expression signatures in SLE which may be true or may not be true. If the study was fully reported, well controlled and designed, as we have

to assume that the “much larger unpublished study” might be, then maybe these observations could be supported. It is unfortunate that the authors have not elected to await the publication of the larger parent study to allow these data on SLE to be properly reviewed and published. I would worry that premature publication of a potentially superficial analysis of these data would be disservice to both the primary study investigators, the patients who provided samples to that study and to the SLE research community at large.

Following this comment, we have removed the data set in question and instead replaced it with two different examples. The first showcases *Vision* on an analysis of PBMCs from SLE patients, published by Kang, Subramaniam, and Targ et al. (Nature Biotech 2017) with the more traditional use of PCA to form a latent space. The second uses *Vision* to analyze single-cell AML data from Galen et al. (Cell 2019), instead using a variational autoencoder-derived latent space with scVI. We believe that these two examples, in addition to the existing example in hematopoiesis, serve to showcase *Vision* in a variety of experimental situations and demonstrate its flexibility with regard to use of latent models of single-cell variability.

Live versions of the VISION output reports are available at the following links:

- Lupus Stimulated CD4 T cells (Kang, Subramaniam, and Targ et al., Nature Biotech 2017)
- AML Monocytes (Galen et al., Cell 2019)
- Hematopoiesis (Tusi et al., Nature 2018)
- Cite-seq PBMCs (Stoeckius et al., Nature Biotech 2017)

1. Schafflick D, Xu CA, Hartlehnert M, Cole M, Lautwein T, Schulte-Mecklenbeck A, et al. Integrated single cell analysis of blood and cerebrospinal fluid leukocytes in multiple sclerosis [Internet]. bioRxiv. 2019. p. 403527. doi:10.1101/403527
2. van Dijk D, Sharma R, Nainys J, Yim K, Kathail P, Carr AJ, et al. Recovering Gene Interactions from Single-Cell Data Using Data Diffusion. Cell. 2018;174: 716–729.e27.
3. Xu C, Lopez R, Mehlman E, Regier J, Jordan MI, Yosef N. Harmonization and Annotation of Single-cell Transcriptomics data with Deep Generative Models [Internet]. bioRxiv. 2019. p. 532895. doi:10.1101/532895

REVIEWERS' COMMENTS:

Reviewer #1 (Remarks to the Author):

The authors have satisfactorily addressed all of my previous comments. I have no additional concerns.

Reviewer #2 (Remarks to the Author):

The changes the authors have made regarding the use of the pre-publication SLE dataset are acceptable and actually make this manuscript more focused on the comparison of this tool and approach in a systems immunological analysis plan. It provides context of this tool in comparison to previous analyses.